# Occupational Health and Safety Among Brazilian Immigrant Women in the United States: A Cross-Sectional Survey

**DOI:** 10.3390/ijerph22060963

**Published:** 2025-06-19

**Authors:** Ashley Scott, Salima F. Taylor, Jennifer D. Allen

**Affiliations:** 1Department of Epidemiology, Boston University School of Public Health, 715 Albany St., Boston, MA 20118, USA; aascott@bu.edu; 2Friedman School of Nutrition Science and Policy, Tufts University, Boston, MA 02111, USA; salima.taylor@tufts.edu; 3Department of Community Health, Tufts University, 574 Boston Avenue, Medford, MA 02155, USA

**Keywords:** occupational health and safety, immigrants, Brazilian, women’s health

## Abstract

The Brazilian population in the United States is growing, and many Brazilian workers are employed in settings that may lack occupational health and safety (OHS) protections. In this study, we examined two domains of OHS (measured by the Occupational Health and Safety Vulnerability Survey), namely, Workplace Hazards (potential dangers that may result in injury or illness) and Workplace Vulnerability (inadequate occupational health and safety resources), and described health and demographic characteristics associated with these conditions. Eligible participants were women aged 18 and over, born in Brazil, currently residing in the United States, and employed. A cross-sectional online survey was conducted between July and August 2020. Recruitment occurred through community partnerships and social media. Multivariable models among n = 191 women revealed that greater exposure to Workplace Hazards was associated with employment in private household settings, including childcare and housecleaning (*p* < 0.001). The association between Workplace Vulnerability and jobs in private household services approached statistical significance (*p* = 0.07). Both Workplace Hazards and Workplace Vulnerability were associated with lower incomes and educational attainment, as well as having public insurance. Our findings suggest the need for stronger OHS protections and policies, particularly among those working in private household services, to ensure safer working conditions for Brazilian immigrant women.

## 1. Introduction

Brazilian migration has increased significantly over the past two decades, primarily due to the country’s economic and political turmoil [1]. While initial waves of immigration to the U.S. mainly consisted of single men, there has been a recent rise in the number of families making the journey [2,3]. Once in the U.S., many Brazilians assume insecure employment, characterized by high-risk, low-wage jobs [4]. The fact that an estimated 71% of Brazilians living in the U.S. are undocumented [5,6] and many have limited English language proficiency often leaves these individuals particularly vulnerable to poor working conditions [7,8,9].

Research has consistently demonstrated that immigrants in the U.S. are at increased risk of occupational health and safety issues, resulting in a variety of adverse physical health outcomes [10,11,12,13]. However, most studies of working conditions among immigrants do not disaggregate data by ethnicity and often categorize Brazilians as Hispanic, despite the distinct language, culture, and historical context that differs from other Hispanic subgroups [6,14].

Although Brazilian women in the U.S. have high participation in the labor force, they are understudied. Two-thirds of Brazilian women are employed [9], and 39% hold service jobs, often in housecleaning or childcare [15], compared to employed Brazilian men (12.5%) [2]. Many of these jobs are informal and, as a result, lack oversight or regulatory measures, exposing women to ergonomic hazards (e.g., heavy lifting, rapid work pace without adequate breaks) [16,17] and chemical exposures from cleaning products and vapors [16,18,19]. These conditions pose health and safety risks [16,17,20].

Recent occupational health research has continued to highlight vulnerabilities faced by immigrant women. The COVID-19 pandemic exacerbated workplace inequities for marginalized workers as immigrant women disproportionately held frontline jobs and were exposed to increased health risks [21]. Since the pandemic, researchers have underscored the need for an intersectional approach to occupational health, which considers how immigration status, gender, race, and occupation jointly influence workplace health and safety [22]. This highlights the importance of understanding and addressing the occupational health and safety challenges faced by immigrant women.

Occupational health and safety (OHS) includes both the prevention of illness and injury, as well as the protection of worker health. In this study, we examine two OHS domains: Workplace Hazards and Workplace Vulnerabilities. Workplace Hazards are potential dangers in the work environment that may result in illness or harm from injury [23]. Workplace Vulnerability refers to settings with inadequate safety policies, procedures, and training, as well as lower worker awareness of their rights and responsibilities. Both Workplace Hazards and Vulnerability place workers at an increased risk of physical and mental harm [24]. The goal of this study was to describe Workplace Hazards and Vulnerability among employed Brazilian immigrant women and to examine associations with health and sociodemographic characteristics.

## 2. Materials and Methods

We conducted a cross-sectional online survey between July and August 2020, recruiting women aged 18 years or older who were born in Brazil and currently living in the U.S. Since there is no existing sampling frame for this population, it was not possible to execute probability sampling. Instead, we conducted convenience sampling, recruiting participants through local groups and advocacy organizations serving the Brazilian community, as well as via social media (Facebook and WhatsApp groups). Those interested in participating accessed a link to the study, where they could learn about the study procedures and provide informed consent before completing the survey. Participants could choose to complete the online survey in either English or Portuguese. On average, the survey took 18.5 min to complete. Participants received a link upon completion to provide their contact information in exchange for a USD 20 Amazon gift card. 

### 2.1. Measures

We utilized standardized questions to assess occupational health and safety, as well as sociodemographic characteristics (described below). Best practices for survey translation were used in this process [25]. First, the survey was forward-translated by a native Brazilian Portuguese speaker certified by the American Translators Association. Following forward translation, it was back-translated by three Brazilian-born, Brazilian Portuguese-speaking research team members to ensure linguistic and cultural appropriateness. Next, the survey was pre-tested among five Brazilian immigrant women to assess item flow and comprehension. Feedback from pre-testing indicated that all survey items were clear and required no modifications.

We assessed OHS using eight items from the Occupational Health and Safety Vulnerability Measure developed by the Institute for Work and Health [26]. Specifically, Workplace Hazards evaluate the presence and frequency of physical, chemical, or ergonomic risks. Items inquire about the frequency with which workers are required to “manually lift, carry or push items heavier than 20 kg at least ten times a day,” “do repetitive movements with [your] hands or wrists,” and “interact with hazardous substances.” Participants reported frequency of occurrence and work requirements as “never,” “once a year,” “every 6 months,” “every 3 months,” “every month,” “every week,” or “every day.” A summative score was calculated, such that higher scores reflected more frequent exposures (range 0–3). In the analysis, we collapsed this into two categories: 1) every day and every week, and 2) less than every week. Workers were considered exposed to “high Workplace Hazards” if they experienced a hazard weekly or more often. The internal reliability of these items was good (Cronbach’s alpha = 0.72).

For Workplace Vulnerability, we included questions regarding workplace policies and procedures, awareness of rights and responsibilities, and worker empowerment (i.e., ability to advocate for themselves). Regarding policies and procedures, participants were asked to indicate their agreement or disagreement with the following statements: “Everyone receives the necessary workplace health and safety training,” and “Systems are in place to identify, prevent, and deal with hazards.” To assess worker awareness, we asked participants to indicate the extent of their agreement with the following statements: “I am clear about my rights and responsibilities regarding workplace health and safety,” and “I know what the necessary precautions are that I should take while performing my job.” We presented one statement about worker empowerment: “I feel free to voice concerns or make suggestions about workplace health and safety.” Each item was reported on a scale from strongly agree to strongly disagree. We combined ‘strongly agree’ and ‘agree’ as one category, while the responses ‘not sure/neutral,’ ‘disagree,’ and ‘strongly disagree’ were collapsed as a second category. A summative score (range 0–5) was calculated, with higher scores indicating greater vulnerability. The internal reliability of these items was acceptable (Cronbach’s alpha = 0.60) [27].

We used items from the Brazilian census [28] to assess sociodemographic characteristics, including race and ethnicity (categorized as White, Black, Pardo (mixed race), Indigenous, Multiracial, and another race (including Asian)). Educational attainment was classified into three categories: “complete primary and incomplete secondary,” “complete secondary and incomplete tertiary,” and “complete tertiary.” We collected information about age (continuous years) and household income (<$25,000, $25,001–$50,000, $50,001–$75,000, $75,001–$100,000, and > $100,000; all amounts in USD). Questions to assess health insurance status (yes/no) and health insurance type (public/private/don’t know) were taken from the Behavioral Risk Factor Surveillance System or BRFSS [29]. We also asked if participants had a primary care provider (yes/no), were married/living as married or not married, and the number of years they lived in the U.S. We asked how many hours participants typically work per week (1–10, 10–20, 20–30, 30–40, 40–50, 50–60, 60+). Workplace questions addressed occupation type (private household services/administrator, manager/teacher, other professional/administrative support/sales, retail/or other). Occupation type was dichotomized into “private household services” and “other occupations.” We also collected employment type (employed for wages/self-employed), self-perceived overall health (excellent, good, fair, poor), and languages spoken at home or with friends (Portuguese only, English only, some Portuguese and English, other languages).

### 2.2. Analysis

A total of 446 Brazilian-born women initiated the online survey. We excluded participants with more than 70% missing responses across key variables related to occupational health and safety (n = 64) and those who were not currently employed (n = 111), resulting in a final analytic sample of 271 participants. We retained all 271 participants in the descriptive analyses to provide a comprehensive picture of response patterns and sample characteristics. For subsequent inferential analyses, we employed listwise deletion to handle missing data. Participants were included only if they had complete data on all variables included in the respective regression models. As a result, the analytic sample sizes for multivariable models are smaller, reflecting case-wise exclusions due to partial missingness on one or more predictor variables. We chose listwise deletion due to the relatively low proportion of missing data per variable and the assumption that the data were missing at random (MAR). This approach is consistent with practices in similar survey-based cross-sectional analyses [30].

Descriptive analyses were performed to examine sociodemographic and health characteristics of the sample. Data are presented as means and standard deviations for continuous variables and percentages for categorical variables. In addition to reviewing the frequency and percentage of responses on individual items, we calculated means and standard deviations for total Worksite Hazards and Vulnerability scores. Worksite Hazard scores (range 0–3) were dichotomized to represent “low” (0–1) or “high” hazards (2–3), and Worksite Vulnerability scores ranged from 0 to 5. For each scale, measures of association with sociodemographic characteristics were completed using linear regression for continuous variables and Pearson correlation for categorical variables. A multivariable linear regression model was used to assess Workplace Hazards and Workplace Vulnerability, controlling for significant sociodemographic characteristics that were significant at the *p*-value of <0.10. Data are presented as beta coefficients (B) at a 95% level of significance (*p*-value < 0.05). Model fit is presented as R^2^, and confidence intervals (CI) are included in the multivariable regression models. All data analysis was conducted using STATA version SE [31].

## 3. Results

### 3.1. Sample Characteristics

Descriptive statistics are presented in Table 1. The mean age was 23 years (SD = 11), and the mean number of years living in the U.S. was 13 years (SD = 9). The majority identified their race as White (59%), with 23% identifying as Pardo (mixed-race). More than two-thirds (69%) were married or living as married, and 46% had household incomes of $50,000 or below. Approximately 48% had completed tertiary education (U.S. college degree equivalent). Most (81%) had health insurance, with more than one-third (35%) having public insurance. Most (67%) respondents worked more than 40 h a week and were employed for wages (54%) or were self-employed (48%). Forty four percent were employed in private households.

### 3.2. Occupational Health and Safety

The mean Workplace Hazards score was 1.0 (SD 1.0). Most participants (80%) reported lifting heavy materials at work less than once a week, compared to every day or every week. More than half (59%) of participants engaged in repetitive movements at work on a daily or weekly basis. Most respondents (65%) reported infrequent interaction with hazardous materials at work. Most (86%) agreed or strongly agreed that they knew precautions to take at work if necessary. Mean scores were significantly associated with years living in the U.S. (*p* = 0.02), racial identity (*p* = 0.009), annual household income (*p* = 0.002), educational level (*p* = 0.03), employment type (*p* = 0.04), occupation (*p* < 0.001), insurance type (*p* < 0.001), self-perceived health (*p* = 0.004), and languages spoken at home (*p* = 0.007) and with friends (*p* = 0.001) (see Table 2).

The mean Workplace Vulnerability score was 1.6 (SD 1.7). Almost half (44%) of participants strongly disagreed, disagreed, or were neutral about having received workplace health and safety training. Most (57%) respondents either agreed or strongly agreed that there were systems in place at work to identify hazards. Clear rights and responsibilities for health and safety at work were in place for 73% of participants. Mean scores were significantly associated with having health insurance (*p* = 0.003), employment type (*p* = 0.01), and occupation (*p* = 0.03). Languages spoken at home were related to Workplace Vulnerability at the *p* = 0.1 level (see Table 3).

### 3.3. Multivariable Analyses

In multivariable linear regression, the model explained approximately a third of the variance (R^2^ = 0.34), indicating that sociodemographic characteristics accounted for a moderate proportion of the variability in Workplace Hazards scores. Workplace Hazard scores were significantly associated with household income, health insurance type, languages spoken at home and with friends, and occupations in private household services. Compared to women making less than $25,000, women making between $75,001 and $100,000 had a 0.5-unit higher Worksite Hazard score. Compared to women with public insurance, having private insurance was associated with a 0.46-unit higher score in Workplace Hazards. Being employed in private households, compared to other occupations, was associated with a 0.7-unit lower score in exposure to Worksite Hazards. Speaking other languages at home, compared to speaking Portuguese, was associated with a 1.6-unit lower score in Worksite Hazards. Speaking some English and Portuguese with friends, compared to speaking Portuguese only, was associated with a 0.3-unit lower level of exposure to Workplace Hazards. Reporting excellent health, compared to poor health, was marginally associated with a 0.3 unit decrease in exposure to Worksite Hazards (see Table 4).

The Workplace Vulnerability multivariable linear regression model explained 8.4% of the variance in Workplace Vulnerability scores (R^2^ = 0.084). Employment type was significantly associated with Workplace Vulnerability (Table 5). Being self-employed, compared to being employed for wages, was associated with a 0.6-unit higher Worksite Vulnerability score. Being employed in private household services was marginally associated with a 0.4-unit higher Workplace Vulnerability score (see Table 5).

## 4. Discussion

Our study contributes to the sparse literature on OHS among Brazilian women working in the U.S. We employed a new dataset to document critical domains of occupational health and safety—both ergonomic risks and chemical exposures—among a population that has received little attention in the research on workplace health and safety to date. Our findings reveal important insights into the risks faced by this population, particularly those employed in informal work settings, such as childcare and housecleaning. We also demonstrate the link between lower income and educational attainment and risks, which underscores the need for targeted interventions to improve workplace protections for women who experience social disadvantage beyond immigration status.

We found that women working in private household settings were at greater risk of exposure to ergonomic and chemical hazards. Furthermore, they were less likely to have received training on workplace health and safety measures and were less informed about their rights in the workplace. Our findings are generally consistent with the available literature on occupational and safety issues among Brazilian workers in the U.S., the majority of which was conducted over a decade ago. In 2012, Siqueira and Jansen conducted a study of over 500 Brazilian immigrant workers in Eastern Massachusetts, most of whom were employed in the construction, housekeeping, and food services sectors. They found that a large proportion were exposed to chemical, physical, and psychosocial hazards. Many workers lacked adequate training on workplace safety and reported limited access to protective equipment. Work-related injuries and illnesses were common, but many did not report them due to fear of job loss or immigration concerns [16]. Similar issues were identified in a smaller sample (n = 50) of Brazilian immigrant housecleaners, as reported by Siqueira and Roche in 2013 [17]. A more recent study conducted in 2016 that included 198 Brazilian (predominantly) domestic women workers found that those with low English language and non-legalized status reported poor working environments compared to those with documented legal status (55.6% vs. 34.3%). Working conditions included less access to personal protective equipment and difficulty negotiating pay and contracts due to low English proficiency [32]. Other quantitative [19] and qualitative studies [18,33,34] have emphasized similar results. These studies highlight the importance of concerns about training gaps in workplace safety, a lack of protective equipment, and exposure to hazards among Brazilian immigrant women. 

Before discussing the study’s implications, we acknowledge its limitations. First, this was a convenience sample, which may have resulted in selection bias. We were not able to conduct probability sampling, as there is no available sampling frame for Brazilian immigrant women. Therefore, results must be interpreted with appropriate caution. Theoretically, there is potential for bias in either direction. It is possible that women who completed the survey were more likely to be aware of workplace risks, which could result in an overestimate of the issues we observed. Alternatively, there could be an underestimate of risks as workers generally underestimate job risk and their ability to self-protect from harm [16]. Brazilian women may have been willing to accept more hazardous jobs with inadequate protections during the COVID-19 pandemic to relieve their family’s economic insecurity. The consistency of our findings with prior studies provides some reassurance about the potential bias in the observed relationships. Moreover, even though we utilized standard best practices for cultural and linguistic survey translation, there remains potential for bias introduced by the use of survey items developed initially in the English language. Future research should address the preceding limitations and questions that our study cannot.

Despite these limitations, our findings suggest that concerted efforts are needed to improve the working conditions of Brazilian immigrant women. A combination of workplace health and safety protocols, employee training, and active engagement in health and safety initiatives is needed to reduce workers’ vulnerability and exposure to hazards that could lead to illness and injury. This will likely require interventions at multiple levels of the socioecological model, including at the individual, interpersonal, community, and policy levels [35].

At the individual level, training at the workplace can be effective in promoting knowledge, awareness, and practices [36]. However, this will be challenging for this population, as many are doing domestic work. Further, nearly half of the women in our sample spoke only Portuguese. Thus, there is a need to develop centralized community locations for training and to ensure that training is culturally and linguistically appropriate, as well as tailored to different types of work. Alternatively, given the high rates of social media use in the population, e-training could be effective [37]. Furthermore, such interventions are flexible, cost-effective, and can boost workers’ knowledge and skills [37]. However, interventions focusing solely on the individual level put the responsibility for OHS on the worker rather than the employer.

At the employer level, interventions should focus on improving work environments, establishing realistic expectations, providing worksite safety training, and enhancing surveillance and cooperation with regulatory authorities [38]. A review of existing studies of organization-level interventions finds evidence supporting the effectiveness of this approach in more traditional work settings [39]. However, we were unable to locate studies on the effectiveness of employer-level interventions for domestic workers. Reports of exploitation among Brazilian women in domestic worker roles underscore the need for additional interventions to attain these goals [40].

Community-level interventions can also play a crucial role in establishing systems that provide comprehensive support for immigrant communities. Investment in local immigrant-led organizations and collaboratives for workers’ rights can and do provide guidance and support for advocacy and education for immigrant women. An excellent example is the Grupo Mulher Brasileira (Brazilian Women’s Group (http://verdeamarelo.org/bwg/, accessed on 10 June 2025). The center originated in 1995 as a non-profit organization dedicated to supporting immigrant workers, primarily Brazilians, in the Greater Boston area, focusing on advocating for their labor and immigration rights. Its mission is to empower immigrants with knowledge about their workplace rights and promote social justice through education and organizing efforts. Additionally, the Vida Verde Women’s Co-Op (https://verdeamarelo.org/vidaverde/, accessed on 10 June 2025), also in the Boston area, supports safe working conditions and education on Workplace Hazards often faced by Brazilian immigrant women [41]. Both groups were part of a coalition that brought the Domestic Workers Bill of Rights to Massachusetts in 2015 [42,43]. Collaborative initiatives led by immigrant organizations should be fully funded and expanded to conduct this vital work. While this study was conducted in Massachusetts with these extraordinary organizations, our findings suggest that more should be done.

Interventions on the societal and policy levels are also essential. Fundamental efforts are required to combat anti-immigrant xenophobia and racism in the U.S., which is rapidly escalating in the U.S. [44]. Murray and colleagues provide an excellent review of the necessary work to address xenophobia and discrimination against immigrants [45]. Policymakers need to address the exploitation of immigrant workers by instituting and enforcing worker protections and rights. Social, economic, and labor policies on occupational health at the state and federal levels lack sufficient protections for immigrant workers, including OSHA coverage in private homes, increased federal minimum wage laws, and legal protections for immigrant workers regardless of immigration status [4].

## 5. Conclusions

This study demonstrates that Brazilian immigrant women may experience significant health threats because of OHS conditions associated with their employment, which may be further compounded by socioeconomic factors. These results underscore the need for enhanced OHS regulations and oversight, particularly for workers in informal work settings. Further research and policy change are needed to mitigate workplace risks and ensure safer and supported working conditions for immigrant women.

## Figures and Tables

**Table 1 ijerph-22-00963-t001:** Sociodemographic characteristics and self-perceived health, n = 271, Brazilian Women’s Health Study.

Sample Characteristics	Mean	SD
Age in years	23	11
Years in the U.S.	13	9
	**n**	**%**
Racial identity		
Black	17	6
Indigenous	3	1
Multiracial	11	4
Another race	14	6
Pardo	63	23
White	159	59
Married/living as married	186	69
Household income		
<$25,000 (USD)	67	25
$25,001–$50,000 (USD)	58	21
$50,001–$75,000 (USD)	48	18
$75,001–$100,000 (USD)	37	14
>$100,001 (USD)	41	15
Don’t know	20	7
Educational level		
Complete primary education and incomplete secondary education	50	19
Complete secondary and incomplete tertiary education	90	33
Complete tertiary education	129	48
Don’t know	1	0.4
Missing	1	0.4
Occupation		
Private household services (e.g., housecleaner, childcare)	102	44
Other occupations *	131	56
Missing	38	14
Employment type		
Employed for wages	125	52
Self-employed	114	48
Missing	32	12
Health insurance		
Yes	215	81
No	44	17
Don’t know	7	3
Missing	5	2
Health insurance type		
Public	96	44
Private	109	50
Don’t know	15	7
Missing	51	19
Number of hours worked		
<20 h	39	17
20–39	40	17
≥40 h	158	67
Missing	34	13
Self-perceived health		
Poor	1	0.4
Fair	25	9
Good	162	61
Excellent	78	29
Missing	5	1.9
Languages spoken at home		
Portuguese only	119	46
English only	25	9
Some English and Portuguese	124	46
Another language	3	1
Languages spoken with friends		
Portuguese only	82	30
English only	4	2
Some English and Portuguese	183	68
Another language	2	1

Totals may not sum to 100% due to rounding. * Other occupations included administrator (manager), n = 14; teacher, n = 19; professional, n = 42; administrative support (clerical), n = 19; sales (retail), n = 12; other, n = 25.

**Table 2 ijerph-22-00963-t002:** Workplace Hazards scores by health and sociodemographic characteristics, n = 228 **^§^**, Brazilian Women’s Health Study.

	Mean	SD	
Total Workplace Hazard Scores	1.0	1.0	
Workplace Hazard Scores (range 0–3)	**Low Hazards** (score 0–1)	**High Hazards** (score 2–3)	** *p* ** **-value ***
Sample size	(n = 167)	(n = 61)	
Age in years (mean, SD)	23 (11)	23 (9)	0.9
Years in the U.S.	14 (9)	11 (8)	**0.02**
	n (%)	n (%)	
Racial identity			**0.009**
Black	10 (6)	4 (7)	
Multiracial	7 (4)	4 (7)	
Indigenous	2 (1)	1 (2)	
Another race	10 (6)	3 (5)	
Pardo	26 (16)	23 (38)	
White	112 (67)	26 (43)	
Marital status			0.7
Unmarried	48 (29)	19 (31)	
Married/living as married	119 (71)	42 (69)	
Annual household income			**0.002**
<$25,000 (USD)	32 (19)	22 (36)	
$25,001–$50,000 (USD)	33 (20)	19 (31)	
$50,001–$75,000 (USD)	35 (21)	7 (12)	
$75,001–$100,000 (USD)	26 (16)	6 (10)	
>$100,001 (USD)	32 (19)	2 (3)	
Don’t know/missing	9 (5)	5 (8)	
Educational level			**0.03**
Complete primary education and incomplete secondary education	23 (14)	16 (26)	
Complete secondary and incomplete tertiary education	48 (29)	22 (36)	
Complete tertiary education	95 (57)	23 (38)	
Don’t know/missing	1 (1)	0 (0)	
Employment type			**0.04**
Employed for wages	94 (56)	25 (41)	
Self-employed	73 (44)	36 (59)	
Occupation			**<0.001**
Private household services	46 (28)	52 (85)	
Other occupations **	121 (73)	9 (15)	
Number of hours worked			0.9
<20 h	28 (17)	10 (16)	
20 to 30 h	26 (16)	11 (18)	
>40 h	112 (68)	40 (66)	
Health insurance			0.3
No	26 (16)	11 (18)	
Yes	135 (81)	50 (82)	
Don’t know/missing	6 (4)	0 (0)	
Insurance type			**<0.001**
Public	48 (34)	35 (70)	
Private	84 (60)	11 (22)	
Don’t know	9 (6)	4 (8)	
Self-perceived health			**0.004**
Excellent	57 (34)	13 (21)	
Fair	9 (5)	11 (18)	
Good	101 (61)	36 (59)	
Poor	0 (0)	1 (2)	
Languages spoken at home			**0.007**
Portuguese only	62 (37)	38 (62)	
English only	18 (11)	3 (5)	
Some English and Portuguese	84 (50)	20 (33)	
Another language	3 (2)	0 (0)	
Languages spoken with friends			**0.001**
Portuguese only	36 (22)	29 (48)	
English only	2 (1)	1 (2)	
Some English and Portuguese	127 (76)	31 (51)	
Other language	2 (1)	0 (0)	

**^§^** Sample size reduced from 271 to 228 due to missing data. * Linear regression was used for continuous variables. Pearson correlation was used for categorical variables. ** Other occupations: administrator (manager), n = 14; teacher, n = 19; professional, n = 42; administrative support (clerical), n = 19; sales (retail), n = 12; other, n = 25.

**Table 3 ijerph-22-00963-t003:** Workplace Vulnerability scores by health and socio-demographic characteristics, n = 227 ^§^, Brazilian Health Women’s Study.

	Mean				(SD)		
Total Workplace Vulnerability Score	1.6				(1.7)		
**Workplace Vulnerability Scores (range 0–5)**							
	**0**	**1**	**2**	**3**	**4**	**5**	** *p* ** **-value ***
Sample size	(n = 93)	(n = 42)	(n = 16)	(n = 39)	(n = 17)	(n = 20)	
	**Mean (SD)**						
Age (mean, SD)	23 (11)	25 (12)	19 (11)	24 (9.6)	23 (9)	20 (9)	0.4
Years in US (mean, SD)	13 (8)	13 (10)	9 (7)	14 (11)	11 (8)	11 (9)	0.4
	**n (%)**						
Racial identity							0.3
White	63 (68)	26 (62)	8 (50)	23 (59)	6 (35)	10 (50)	
Black	8 (9)	3 (7)	1 (6)	0 (0)	2 (12)	0 (0)	
Multiracial	5 (5)	3 (7)	0 (0)	1 (3)	0 (0)	2 (10)	
Indigenous	1 (1)	1 (2)	0 (0)	0 (0)	0 (0)	1 (5)	
Pardo	12 (13)	7 (17)	5 (31)	13 (33)	7 (41)	6 (30)	
Another race	4 (4)	2 (5)	2 (13)	2 (5)	2 (12)	1 (5)	
Marital status							0.9
Unmarried	31 (33)	10 (24)	4 (25)	12 (31)	5 (29)	7 (35)	
Married/living as married	62 (67)	32 (76)	12 (75)	27 (69)	12 (71)	13 (65)	
Annual household income							0.2
<$25,000 (USD)	22 (24)	8 (19)	3 (19)	10 (26)	4 (24)	8 (40)	
$25,001–$50,000 (USD)	23 (25)	5 (12)	3 (19)	9 (23)	7 (41)	5 (25)	
$50,001–$75,000 (USD)	21 (23)	6 (14)	4 (25)	6 (15)	2 (12)	3 (15)	
$75,001–$100,000 (USD)	8 (9)	12 (29)	1 (6)	7 (18)	2 (12)	1 (5)	
>$100,001 (USD)	14 (15)	9 (21)	2 (13)	6 (15)	2 (12)	1 (5)	
Don’t know	5 (5)	2 (5)	3 (19)	1 (3)	0 (0)	2 (10)	
Educational level							0.5
Complete primary education and incomplete secondary education	13 (14)	3 (7)	4 (25)	9 (23)	2 (12)	7 (35)	
Complete secondary and incomplete tertiary education	25 (27)	16 (38)	4 (25)	13 (33)	6 (35)	6 (30)	
Complete tertiary education	54 (58)	23 (55)	8 (50)	17 (44)	9 (53)	7 (35)	
Health insurance							
No	13 (14)	6 (14)	1 (6)	9 (23)	5 (29)	4 (20)	**0.003**
Yes	79 (85)	35 (83)	12 (75)	30 (77)	12 (71)	16 (80)	
Don’t know	1 (1)	1 (2)	3 (19)	0 (0)	0 (0)	0 (0)	
Insurance type							0.2
Public	30 (38)	11 (31)	10 (67)	16 (53)	8 (67)	7 (44)	
Private	44 (55)	24 (67)	4 (27)	12 (40)	4 (33)	7 (44)	
Don’t know	6 (8)	1 (3)	1 (7)	2 (7)	0 (0)	2 (13)	
Number of hours worked							
<20 h	18 (20)	5 (12)	4 (25)	5 (13)	2 (12)	3 (15)	0.3
20 to 30 h	14 (15)	7 (17)	0 (0)	10 (26)	1 (6)	6 (30)	
≥40 h	60 (65)	30 (71)	12 (75)	24 (62)	14 (82)	11 (55)	
Employment type							**0.01**
Employed for wages	58 (62)	26 (62)	7 (44)	14 (36)	7 (41)	6 (30)	
Self-employed	35 (38)	16 (38)	9 (56)	25 (64)	10 (59)	14 (70)	
Occupation							**0.03**
Private household services	29 (31)	19 (45)	9 (56)	18 (46)	11 (65)	12 (60)	
Other occupations **	64 (69)	23 (55)	7 (44)	21 (54)	6 (35)	8 (40)	
Self-perceived health							0.4
Excellent	31 (33)	13 (31)	5 (31)	12 (31)	4 (24)	4 (20)	
Fair	7 (8)	4 (10)	1 (6)	4 (10)	1 (6)	3 (15)	
Good	55 (59)	25 (60)	10 (63)	23 (59)	11 (65)	13 (65)	
Poor	0 (0)	0 (0)	0 (0)	0 (0)	1 (6)	0 (0)	
Languages spoken at home							0.1
Portuguese only	37 (40)	16 (38)	11 (69)	16 (41)	10 (59)	10 (50)	
English only	10 (11)	3 (7)	3 (19)	1 (3)	1 (6)	3 (15)	
Some English and Portuguese	46 (50)	22 (52)	2 (13)	20 (51)	6 (35)	7 (35)	
Other language	0 (0)	1 (2)	0 (0)	2 (5)	0 (0)	0 (0)	
Languages spoken with friends							0.8
Portuguese only	24 (26)	12 (29)	5 (31)	15 (39)	3 (18)	6 (30)	
English only	1 (1)	0 (0)	0 (0)	1 (3)	0 (0)	1 (5)	
Some English and Portuguese	67 (72)	30 (71)	11 (69)	22 (56)	14 (82)	13 (65)	
Other language	1 (1)	0 (0)	0 (0)	1 (3)	0 (0)	0 (0)	

**^§^** Sample size reduced from 271 to 227 due to missing data. * Linear regression was used for continuous variables. Pearson correlation was used for categorical variables. ** Other occupations: administrator (manager), n = 14; teacher, n = 19; professional, n = 42; administrative support (clerical), n = 19; sales (retail), n = 12; other, n = 25.

**Table 4 ijerph-22-00963-t004:** Multivariable linear regression model: Workplace Hazards and health and socio-demographic characteristics, n = 191 ^§^, Brazilian Women’s Health Study.

Characteristic	B	*p*-Value *	Confidence Interval (CI)
Years in the U.S.	0.01	0.4	−0.01, 0.02
Educational level			
Complete primary education and incomplete secondary education	--	--	--
Complete secondary and incomplete tertiary education	0.2	0.4	−0.23, 0.57
Complete tertiary education	0.3	0.1	−0.08, 0.77
Don’t know/missing	0.6	0.4	−1.11, 2.34
Racial identity			
White	--	--	--
Black	−0.01	0.9	−0.51, 0.53
Multiracial	0.4	0.2	−0.23, 0.97
Indigenous	−0.7	0.2	−1.68, 0.37
Pardo	0.1	0.5	−0.21, 0.43
Another race	−0.3	0.3	−0.83, 0.29
Household income			
<USD 25,000	--	--	--
<$25,000 (USD)	0.2	0.4	−0.22, 0.55
$25,001–$50,000 (USD)	0.05	0.8	−0.37, 0.48
$50,001–$75,000 (USD)	0.5	**0.05**	0.002, 1.04
$75,001–$100,000 (USD)	0.1	0.6	−0.36, 0.63
Don’t know/missing	−0.5	0.09	−1.04, 0.08
Health insurance type			
Public	--	--	--
Private	−0.4	**0.02**	−0.80, −0.05
Don’t know/missing	−0.1	0.6	−0.67, 0.41
Employment type			
Employed for wages	--	--	--
Self-employed	−0.2	0.2	−0.46, 0.11
Occupation			
Other occupations **	--	--	--
Private household services	0.7	**<0.001**	**0.42, 1.06**
Self-perceived health ***			
Poor	--	--	--
Fair	0.3	0.2	−0.19, 0.76
Excellent	−0.3	0.06	−0.56, 0.01
Languages spoken at home			
Portuguese only	--	--	--
English only	0.07	0.7	−0.41, 0.56
Some English and Portuguese	−0.3	0.8	−0.54, 0.03
Another language	−1.6	**0.01**	−2.81, −0.31
Languages spoken with friends			
Portuguese only	--	--	--
English only	−0.7	0.4	−2.47, 0.99
Some English and Portuguese	−0.4	**0.03**	−0.70, −0.02

**^§^** Sample size reduced from 271 to 191 due to missing data. * Linear regression was used for continuous variables. Pearson correlation was used for categorical variables. ** Other occupations: administrator (manager), n = 14; teacher, n = 19; professional, n = 42; administrative support (clerical), n = 19; sales (retail), n = 12; other, n = 25. *** “Good” self-reported health category excluded due to collinearity with “excellent” self-reported health category. R-squared = 0.34; adjusted R-squared = 0.24; F(26, 164) = 3.39, *p* < 0.001.

**Table 5 ijerph-22-00963-t005:** Multivariable linear regression model: Workplace Vulnerability score and health and socio-demographic characteristics, n = 227 ^§^, Brazilian Women’s Health Study.

Characteristic	B	*p*-Value *	Confidence Interval (CI)
Health insurance			
No	--	--	
Yes	−0.3	0.3	−0.92, 0.27
Don’t know/not sure	−0.5	0.5	−2.08, 1.04
Employment type			
Employed for wages	--	--	
Self-employed	0.6	**0.01**	**0.11, 1.08**
Occupation			
Other occupations **	--	--	
Private household services (e.g., housecleaner, childcare)	0.4	0.07	−0.06, 0.91
Languages spoken at home			
Portuguese only	--	--	
English only	0.02	0.9	−0.78, 0.83
Some English and Portuguese	−0.2	0.4	−0.66, 0.28
Another language	0.9	0.4	−1.06, 2.78

**^§^** Sample size reduced from 271 to 227 due to missing data. * Linear regression was used for continuous variables, age and year in the U.S.; Pearson correlation was used for categorical variables, income, education, insurance, insurance type, hours, employment, occupation, and perceived health. ** Other occupations: administrator (manager), n = 14; teacher, n = 19; professional, n = 42; administrative support (clerical), n = 19; sales (retail), n = 12; other, n = 25. R-squared = 0.084; adjusted R-squared = 0.055; F(7, 219) = 2.87, *p* = 0.007.

## Data Availability

The raw data supporting the conclusions of this article will be made available by the authors on request.

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
