# Peer review of "Occupational Health and Safety Among Brazilian Immigrant Women in the United States: A Cross-Sectional Survey"

_ijerph, 2025, doi:10.3390/ijerph22060963_

Round 1
Reviewer 1 Report
Comments and Suggestions for Authors
Congratulations to the authors on submitting this manuscript focused on occupational health disparities among a vulnerable and under-researched population: Brazilian immigrant women in informal labor sectors. The following comments are intended to support the authors in improving the manuscript for potential publication.
The abstract includes no quantitative results — no hazard or vulnerability scores, no beta coefficients, no p-values. This makes the findings seem ambiguous or superficial. The final sentence ("Our findings highlight the need...") is vague and lacks impact. It should more clearly convey the study’s practical or policy implications.
The Introduction section does not clearly develop a coherent argument. It jumps from general facts about immigration to vulnerability without presenting a coherent framework for the argument that would naturally lead to the purpose for the study. Reorganize the introduction to guide the reader through a logical progression — beginning with the general context, then identifying specific gaps in knowledge, highlighting unanswered questions or inconsistencies in the literature, articulating the rationale for the study, and finally stating the research aims. Highlight conflicts, omissions, or unresolved questions in the existing literature (e.g., limited gendered data, lack of informal labor analysis, linguistic barriers). Consider referencing more recent occupational health research or trends in gig/informal labor post-COVID. The originality of the study is not demonstrated. The study is important — but that’s not the same as showing it’s novel (Does it introduce a new population? New variables? A new theoretical framework? A new dataset?) Clarify the paper’s contribution and relevance to specific stakeholders (e.g., occupational health regulators, immigrant advocacy organizations, state policymakers).
On Methods section the manuscript doesn’t clarify if the survey was pre-tested or piloted (e.g., for clarity, linguistic equivalence) and if the English and Portuguese versions were validated translations or independently back-translated. Consider adding detail on instrument translation, pretesting, or cultural adaptation to enhance reproducibility and transparency. Clarify the approach to missing data — especially given the reduction in analytic sample size (~30% lost: e.g., sample reduced from 271 to 191 or 228).
The Results section addresses the study’s central hypotheses using appropriate methods, including multivariable regression and stratified analysis.
Table 2 the authors could show mean hazard scores by exposure group for easier comparison (e.g., high vs. low risk groups).
Table 4 - the model has reduced sample size (N=191) due to missing data — but the authors don’t report model diagnostics, R², or how missingness was handled.
Table 5 - the model is limited in scope — only 3 predictors are included. It’s unclear why other variables (e.g., income, language, education) weren’t tested here. Consider including a full model with covariates parallel to the hazards model
Confidence intervals are not reported in the regression tables. This is a major weakness.
The Discussion section reports key findings in a balanced way, interpreting results in the context of prior literature on Brazilian immigrant workers and occupational health. The authors do a good job referencing prior work but miss the opportunity to clarify this study’s unique contribution to the field.The limitation section does not mention of how missing data affected the analysis (sample dropped from 446 to 191 in some models). The authors do not acknowledge language translation issues, or the absence of power calculations.
Some sentences are overly long or lack proper structure, and certain sections would benefit from more concise and polished phrasing.
Below are specific examples where clarity and grammar could be improved:
In the abstract: the sentence: “...multilinear regression models showed that those who had more significant exposure to Workplace Hazards and greater Workplace Vulnerability when working in private household services…” — is grammatically incomplete. It reads like a fragment and lacks a proper verb to complete the thought.
On Methods (Lines 63–64): “We restructured these analyses to women employed at the time of the survey.” The phrase is unclear and grammatically awkward.
On Discussion (Lines 224–227): "We found that women working in private household services were more likely to be exposed to ergonomic or chemical risks and were less likely to have workplace health and safety training, be aware of their rights..." - Long sentence with unbalanced structure.
Author Response
Congratulations to the authors on submitting this manuscript focused on occupational health disparities among a vulnerable and under-researched population: Brazilian immigrant women in informal labor sectors. The following comments are intended to support the authors in improving the manuscript for potential publication.
- The abstract includes no quantitative results — no hazard or vulnerability scores, no beta coefficients, no p-values. This makes the findings seem ambiguous or superficial.
Author response: Thank you for this suggestion. The findings have been updated to include the following results:
“Multivariable models among N=191 women revealed that greater exposure to Workplace Hazards was associated with employment in private household settings, including childcare and housecleaning (p<0.001), while the association between Workplace Vulnerability and employment in private household services approached statistical significance (p = 0.07). Both Workplace Hazards and Vulnerability were associated with lower incomes and educational attainment, as well as having public insurance.”
- The final sentence ("Our findings highlight the need...") is vague and lacks impact. It should more clearly convey the study’s practical or policy implications.
Author response: We have updated the final sentence to reflect a need for better policy and protections based on our findings. Specifically, we state:
“Our findings suggest the need for stronger OHS protections and policies, particularly among those working in private household services, to ensure safer working conditions for Brazilian immigrant women. “
- The Introduction section does not clearly develop a coherent argument. It jumps from general facts about immigration to vulnerability without presenting a coherent framework for the argument that would naturally lead to the purpose for the study. Reorganize the introduction to guide the reader through a logical progression — beginning with the general context, then identifying specific gaps in knowledge, highlighting unanswered questions or inconsistencies in the literature, articulating the rationale for the study, and finally stating the research aims. Highlight conflicts, omissions, or unresolved questions in the existing literature (e.g., limited gendered data, lack of informal labor analysis, linguistic barriers). Consider referencing more recent occupational health research or trends in gig/informal labor post-COVID.
Author response: Thank you for your suggestion. We have reorganized the introduction to demonstrate the need for additional research into occupational health and safety among Brazilian women. We highlighted the gaps in knowledge due to prior studies’ data limitations (e.g., Brazilians are combined with Latinos/a) and a lack of research on immigrant women despite their participation in the workforce, particularly in the private household sector.
We appreciate your suggestion of including occupational health research in the post-pandemic era, and added more recent publications, including the following:
“Recent occupational health research has continued to highlight vulnerabilities faced by immigrant women. The COVID-19 pandemic exacerbated workplace inequities for marginalized workers as immigrant women disproportionately held frontline jobs and were exposed to increased health risks ​[21]​. Since the pandemic, researchers have underscored the need for an intersectional approach to occupational health, which considers how immigration status, gender, race, and occupation jointly influence workplace health and safety ​[22]​. This highlights the importance of understanding and addressing the occupational health and safety challenges faced by immigrant women.”
- The originality of the study is not demonstrated. The study is important — but that’s not the same as showing it’s novel (Does it introduce a new population? New variables? A new theoretical framework? A new dataset?) Clarify the paper’s contribution and relevance to specific stakeholders (e.g., occupational health regulators, immigrant advocacy organizations, state policymakers).
Author response: Thank you for this suggestion. We have included further details on how our paper makes a unique contribution to the field. Specifically, we include the following in the discussion section:
“Our study contributes to the limited literature on occupational health and safety among Brazilian women working in the United States. We utilize a new dataset to document critical domains of occupational health and safety —both ergonomic risks and chemical exposures —among a population that has received limited attention in the existing research on workplace health and safety. Our findings provide important insights into the risks faced by this population, particularly those employed in informal work settings, such as childcare and housekeeping. We also demonstrate the link between lower income and educational attainment and risks, which underscores the need for targeted interventions to improve workplace protections for women who experience social disadvantage beyond immigration status.”
- On Methods section the manuscript doesn’t clarify if the survey was pre-tested or piloted (e.g., for clarity, linguistic equivalence) and if the English and Portuguese versions were validated translations or independently back translated. Consider adding detail on instrument translation, pretesting, or cultural adaptation to enhance reproducibility and transparency.
Author response: We have added details about the survey development and pre-testing process. Specifically, the process is described as the following:
“We utilized standardized questions to assess occupational health and safety, as well as sociodemographic characteristics (described below). The survey was first translated into Brazilian Portuguese by a certified American Translators Association translator and subsequently reviewed for linguistic and cultural appropriateness by three research team members who were Brazilian-born and fluent in Portuguese. It was subsequently pre-tested among five Brazilian immigrant women to assess item flow and comprehension. Feedback from the pretesting indicated that all survey items were clear and required no modifications.”
- Clarify the approach to missing data — especially given the reduction in analytic sample size (~30% lost: e.g., sample reduced from 271 to 191 or 228).
Author response: Per your suggestion, the following text has been added to clarify the approach used to manage missing data.
“A total of 446 Brazilian-born women initiated the online survey. We excluded participants with more than 70% missing responses across key variables related to occupational health and safety (n = 64) and those who were not currently employed (n = 111), resulting in a final analytic sample of 271 participants. We retained all 271 participants in the descriptive analyses (Table 1) to provide a comprehensive picture of response patterns and sample characteristics. For inferential analyses (Tables 2-5), we employed listwise deletion to handle missing data. Participants were included only if they had complete data on all variables included in the respective regression models. As a result, the analytic sample sizes for multivariable models are smaller (e.g., n = 228), reflecting case-wise exclusions due to partial missingness on one or more predictor variables. The number of observations used in each model is reported in the table footnotes. We chose listwise deletion due to the relatively low proportion of missing data per variable and the assumption that the data were missing at random (MAR). This approach is consistent with practices in similar survey-based cross-sectional analyses [29].”
- The Results section addresses the study’s central hypotheses using appropriate methods, including multivariable regression and stratified analysis.
Author response: Thank you.
- Table 2 the authors could show mean hazard scores by exposure group for easier comparison (e.g., high vs. Low-risk groups).
Author response: Thank you for this suggestion to present mean hazard scores by exposure group (e.g., high vs. low risk) in Table 2 to facilitate comparison. We have modified Table 2 and the subsequent multivariable table to reflect the slightly modified results.
- Table 4 - the model has reduced sample size (N=191) due to missing data — but the authors don’t report model diagnostics, R², or how missingness was handled.
Author response: Per your recommendation, we have included additional explanation of how missing data was handled in the analysis section (see above). We have also included the R2 as a footnote in the Tables and described these in the results section.
- Table 5 - the model is limited in scope — only 3 predictors are included. It’s unclear why other variables (e.g., income, language, education) weren’t tested here. Consider including a full model with covariates parallel to the hazards model
Author response: This is a valid point. However, before conducting the study, it was determined that only the variables with a p-value of less than 0.1 would be included in the regression models. For Workplace Vulnerability (Table 5), variables significant in bivariate models included only insurance type (p=0.003), employment type (p=0.01), occupation (p=0.03), and language (p=0.1). We did not feel it appropriate to change the criteria for adding variables based on p-values after the initial analysis was conducted.
- Confidence intervals are not reported in the regression tables. This is a major weakness.’
Author response: Confidence Intervals have been added to Tables 4 and 5.
- The Discussion section reports key findings in a balanced way, interpreting results in the context of prior literature on Brazilian immigrant workers and occupational health. The authors do a good job referencing prior work but miss the opportunity to clarify this study’s unique contribution to the field.
Author response: Thank you for this suggestion. We have included further details on how our paper makes a unique contribution to the field (see response to comment #4 above)
- The limitation section does not mention of how missing data affected the analysis (sample dropped from 446 to 191 in some models). The authors do not acknowledge language translation issues, or the absence of power calculations.
Author response: We have added additional information about how missing data was handled to the analysis section (see above).
Regarding translation of the survey, as suggested in comment #5, we have included additional information: “We utilized standardized questions (described below) to assess occupational health and safety, as well as sociodemographic characteristics. The survey was first translated into Brazilian Portuguese by a certified American Translators Association translator and then reviewed for linguistic and cultural appropriateness by three research team members who were Brazilian-born and fluent in Portuguese. It was subsequently pre-tested among five Brazilian immigrant women to assess item flow and comprehension. Feedback from the pretesting indicated that all survey items were clear and required no modifications.”
Minor changes
Some sentences are overly long or lack proper structure, and certain sections would benefit from more concise and polished phrasing.
Author response: We have rewritten sections of the paper to make it more concise and polished.
Below are specific examples where clarity and grammar could be improved:
- In the abstract: the sentence: “...multilinear regression models showed that those who had more significant exposure to Workplace Hazards and greater Workplace Vulnerability when working in private household services…” — is grammatically incomplete. It reads like a fragment and lacks a proper verb to complete the thought.
Author response: The sentence has been modified to read “multivariable linear regression models showed that those who had more significant exposure to Workplace Hazards worked in private household services, including childcare, housecleaning (p<0.001). The association between increased Workplace Vulnerability and employment in private household services demonstrated a positive trend and approached statistical significance (p = 0.07).”
- On Methods (Lines 63–64): “We restructured these analyses to women employed at the time of the survey.” The phrase is unclear and grammatically awkward.
Author response: As suggested, we revised this language.
- On Discussion (Lines 224–227): "We found that women working in private household services were more likely to be exposed to ergonomic or chemical risks and were less likely to have workplace health and safety training, be aware of their rights..." - Long sentence with unbalanced structure.
Author response: As suggested, we have modified the wording: “We found that women working in private household settings were at greater risk of exposure to ergonomic and chemical hazards. Furthermore, they were less likely to have received training on workplace health and safety measures and were less informed about their rights in the workplace.”
Reviewer 2 Report
Comments and Suggestions for Authors
The authors have decided to profile the workplace hazards of immigrant workers. However there are few major issues:
- The recruitment is voluntarily through social media and email. Thus it always comes with a significant bias. How did they overcome this.
- There was around 14.3% having missing data up to the tune of 70% they were excluded. But still the Table 1 shows that one or other parameter is missing in some subjects. Why these were not excluded.
- There are discrepancies in presenting the results. For example in Table 1 for the Employment Type variable there are 271 participants including 32 missing. But in Table 2, the total is 228. Even if we include 32 missing, it totals to 260. Where are missing 11. Thus all the Tables need to be rechecked. Ideally all missing data should not come for analysis.
- Sample size for the study is also not mentioned. What was the sampling frame for the study and how it was obtained is also not mentioned.
Author Response
Thank you for your suggestions.
The authors have decided to profile the workplace hazards of immigrant workers. However, there are few major issues:
- The recruitment is voluntarily through social media and email. Thus it always comes with a significant bias. How did they overcome this.
Author response: This is a valid point. We acknowledge that this is a limitation and have reported this in the Discussion section. Specifically, we state:
“Before discussing the study's implications, we must acknowledge its limitations. First, this was a convenience sample, which may have resulted in selection bias. We were not able to conduct probability sampling, as there is no available sampling frame for Brazilian immigrant women. Therefore, results must be interpreted with appropriate caution. Theoretically, there is potential for bias in either direction. It is possible that women who completed the survey were more likely to be aware of health hazards, which could result in an overestimate of the associations we found. Alternatively, there could be an underestimate of exposures or hazards as workers generally underestimate job risk and their ability to self-protect from harm [31]. In particular, Brazilian women may have been willing to accept more hazardous jobs with inadequate protections to relieve their family’s economic insecurity during the COVID-19 pandemic.”
2. There was around 14.3% having missing data up to the tune of 70% they were excluded. But still the Table 1 shows that one or other parameter is missing in some subjects. Why these were not excluded.
Author response: Thank you for your comment regarding our approach to missing data. As noted, participants with more than 70% missing responses across the 24 key variables were excluded from the analysis to maintain data quality and reduce the influence of substantial item nonresponse (n = 64. For the remaining participants (N = 271), some variables still had occasional missing values, which are reflected in Table 1. These cases were retained to preserve overall sample size and statistical power, as the level of missingness per variable was relatively low and not systematically patterned. In the multivariable models, we relied on listwise deletion, which excluded any case with missing data on a variable included in the model. As a result, the effective sample size in each model may vary slightly based on the number of complete cases available for that specific analysis.
We have clarified this approach in the Methods section by stating:
“A total of 446 Brazilian-born women initiated the online survey. We excluded participants with more than 70% missing responses across key variables related to occupational health and safety (n = 64) and those who were not currently employed (n = 111), resulting in a final analytic sample of 271 participants. We retained all 271 participants in the descriptive analyses (Table 1) to provide a comprehensive picture of response patterns and sample characteristics. For inferential analyses (Tables 2-5), we employed listwise deletion to handle missing data. Participants were included only if they had complete data on all variables included in the respective regression models. As a result, the analytic sample sizes for multivariable models are smaller (e.g., n = 228), reflecting case-wise exclusions due to partial missingness on one or more predictor variables. The number of observations used in each model is reported in the table footnotes. We chose listwise deletion due to the relatively low proportion of missing data per variable and the assumption that the data were missing at random (MAR). This approach is consistent with practices in similar survey-based cross-sectional analyses [29].”\
3. There are discrepancies in presenting the results. For example in Table 1 for the Employment Type variable there are 271 participants including 32 missing. But in Table 2, the total is 228. Even if we include 32 missing, it totals to 260. Where are missing 11. Thus all the Tables need to be rechecked. Ideally all missing data should not come for analysis.
Author response: Thank you for pointing out the discrepancies in sample sizes across the tables. We appreciate your careful attention to detail.
The total number of participants who initiated the survey and met eligibility criteria for this analysis was N = 271. In Table 1, we present descriptive statistics for the full analytic sample, including missing data per variable, to provide transparency about response patterns. As noted, some variables (such as Employment Type) include participants with missing responses, which are explicitly listed to show the extent of item-level nonresponse.
In Table 2 and other multivariable regression models, listwise deletion was applied—meaning that only participants with complete data across all included variables were retained in the analysis. This accounts for the reduced sample size (e.g., N = 228) in those models. The additional discrepancy you noted (11 participants beyond the 32 missing on Employment Type) reflects missingness in other variables used in the model, such as insurance type or language use. These cases were excluded due to incomplete data required for multivariable modeling.
We have clarified the analytic approach in the Methods section, including how missing data were handled at both the descriptive and modeling stages.
4. Sample size for the study is also not mentioned. What was the sampling frame for the study and how it was obtained is also not mentioned.
Author response: Details regarding the analytic sample have been added (see above).
To clarify, our study was a descriptive investigation into the occupational experiences of women born in Brazil and currently residing in the U.S. There is no existing sampling frame for this group. Therefore, it was not possible to execute probability sampling.
We acknowledge the potential bias introduced by convenience sampling, including selection bias and limited generalizability. We acknowledge this in the limitations section of the Discussion.
Round 2
Reviewer 1 Report
Comments and Suggestions for Authors
Suggestions for Authors
The authors significantly improved the manuscript with the inclusion of quantitative findings and clearer policy implications in the abstract, a more structured introduction incorporating recent literature, enhanced methodological transparency through added details on survey translation and pretesting, improved table presentation with R² values and confidence intervals, and notable improvements in grammar and language quality.
While these revisions have addressed many key concerns, the Limitation section still require further attention. Although the methods section now explains how missing data and survey translation were handled, the limitations section would benefit from explicitly acknowledging two important points:
- The potential for translation-related biases, which may influence response consistency across languages.
- The absence of power calculations, especially in light of the sample size reduction in multivariable models.
The quality of English has improved significantly in the revised manuscript. The authors have addressed many grammatical issues, clarified previously awkward phrasing, and improved sentence structure across key sections. However, there are still occasional areas where the language could be further refined for clarity, conciseness, and flow.
Author Response
Thank you for your review and suggestions to improve our manuscript. Please find responses to your comments below and as tracked changes in the manuscript.
1. The potential for translation-related biases, which may influence response consistency across languages.
Author response: Several actions were taken to minimize the potential for translation-related issues, following best practices. First, the survey was forward translated by a native Brazilian Portuguese speaker certified by the American Translators Association. Second, following forward translation, it was back-translated for linguistic and cultural appropriateness by three Brazilian-born, native Brazilian Portuguese-speaking research team members. Third, the survey was subsequently pre-tested among five Brazilian immigrant women to assess item flow and comprehension. Nevertheless, we acknowledge the potential for translation-related issues in the limitation section of the Discussion. Specifically, we note that:
“Moreover, despite the fact that we utilized standard best practices for cultural and linguistic survey translation, there remains potential for bias introduced by use of survey items originally developed in the English language.”
2. The absence of power calculations, especially in light of the sample size reduction in multivariable models.
We respectfully disagree that post-hoc power calculations are necessary for this study. This was a descriptive study that did not test hypotheses; we note appropriate limitations, and we do not claim that the findings are generalizable to a larger population.
Reviewer 2 Report
Comments and Suggestions for Authors
The suggestions given are satisfactorily incorporated.
Author Response
The suggestions given are satisfactorily incorporated.
Author response: Thank you